# Disalicylic Acid Provides Effective Control of *Pectobacterium brasiliense*

**DOI:** 10.3390/microorganisms10122516

**Published:** 2022-12-19

**Authors:** Sapir Tuizer, Manoj Pun, Iris Yedidia, Zohar Kerem

**Affiliations:** 1The Robert H. Smith Faculty of Agriculture, Food and Environment, The Hebrew University of Jerusalem, P.O. Box 12, Rehovot 7610001, Israel; 2The Institute of Plant Sciences, Volcani Center, Agricultural Research Organization (ARO), P.O. Box 15159, Rishon Lezion 7505101, Israel

**Keywords:** salicylic acid, bis(2-carboxyphenyl) succinate, *Pectobacterium brasiliense*, quorum sensing, virulence

## Abstract

Bis(2-carboxyphenyl) succinate (disalicylic acid; DSA) is composed of two salicylic acids connected by a succinyl linker. Here, we propose its use as a new, synthetic plant-protection agent. DSA was shown to control *Pectobacterium brasiliense*, an emerging soft-rot pathogen of potato and ornamental crops, at minimal inhibitory concentrations (MIC) lower than those of salicylic acid. Our computational-docking analysis predicted that DSA would inhibit the quorum-sensing (QS) synthase of *P. brasiliense* ExpI more strongly than SA would. In fact, applying DSA to *P. brasiliense* inhibited its biofilm formation, secretion of plant cell wall-degrading enzymes, motility and production of acyl–homoserine lactones (AHL) and, subsequently, impaired its virulence. DSA also inhibited the production of AHL by a QS-negative *Escherichia coli* strain (DH5α) that had been transformed with *P. brasiliense* AHL synthase, as demonstrated by the biosensors *Chromobacterium violaceaum* CV026 and *E. coli* pSB401. Inhibition of the QS machinery appears to be one of the mechanisms by which DSA inhibits specific virulence determinants. A new route is proposed for the synthesis of DSA, which holds greater potential for use as an anti-virulence agent than its precursor SA. Based on these findings, DSA is an excellent candidate for repurposing for new applications.

## 1. Introduction

The rapidly growing global population and climate changes are putting rapidly increasing pressure on agriculture. FAO’s Scientific Review of the Impact of Climate Change on Plant Pests [1] has predicted that the rise in temperatures and changes in weather patterns due to climate change will increase the risk of pests spreading in agricultural and forest ecosystems. Climate change-induced pest dispersal and proliferation threaten food security as a whole. In light of this, there is a growing need for plant-protection products that will not accumulate and pose additional threats to the environment, while effectively preventing, controlling or eradicating pests. Indeed, pesticides have consistently been shown to reduce food losses to pest damage by up to 40% [1].

The plant-signaling molecule salicylic acid (SA) plays an important role in the induced resistance of plants to bacterial and fungal pathogens and has also been shown to exhibit antimicrobial activity against several bacteria [2,3,4,5]. SA is a monohydroxybenzoic acid that is synthesized by plants and which has a hydroxy group at the ortho position [6]. In addition to its roles in plant growth and development and in plant resistance to different types of stress, SA has also been shown to bind and alter the activity of multiple plant proteins, similar to how the functioning of hormones is mediated by receptors [7]. SA and its derivatives have multiple targets in animals (mainly associated with pathological processes), but also in bacteria, where they have been shown to possess antimicrobial, as well as anti-virulence activity [8]. Together, these findings suggest that SA exerts its defense-associated effects in more than one kingdom via a large number of targets [7]. In addition to its regulatory role in plants, SA has also been shown to control bacterial pathogens and biofilm formation, with potential applications in the food and beverage industries, as well as medicine [9].

Recently, it has been suggested that SA may also interfere with QS in several Gram-negative bacteria, including several strains of pectobacteria [10]. In *P. brasiliense*, SA significantly inhibits the expression of the QS genes *expI* and *expr*, the AHL synthase and response regulator of the QS machinery, and also downregulates QS-dependent genes such as *pecS*, *pel* and *peh* [11]. This downregulation is congruent with the low levels of the AHL signal observed following exposure to the compound.

*Pectobacterium* species cause a variety of disease symptoms (i.e., wilt, soft rot and blackleg) on a wide range of monocot and dicot plants. These diseases are responsible for large economic losses in potato storage and production and in the ornamental-plants industry. Over the last several years, *P. brasiliense* has gained attention as a quickly spreading soft-rot pathogen that is responsible for blackleg and soft-rot infections in potato plants around the world [12]. Virulence depends mainly on the synchronized production of an arsenal of plant cell wall-degrading enzymes (PCDWEs) and several other virulence determinants, of which PCWDEs, biofilm formation and motility have been found to be under the control of the QS system [13,14]. QS-dependent regulation of gene expression in Gram-negative bacteria controls a wide variety of phenotypes, including bioluminescence, biofilm formation, drug resistance, the expression of virulence factors and motility. Therefore, the QS system is considered a promising target for inhibition by antimicrobial and anti-virulence compounds, which may potentially repress the expression of virulence genes, without affecting genes that are essential for basic metabolism and growth [15]. Moreover, as an anti-pathogenic mode of action rather than an antibacterial mode of action, QS inhibition may prevent the emergence of drug-resistant bacteria, as it does not impose the same selective pressure as antimicrobial treatments [13]. In *Pectobacterium* species, QS signaling mainly (but not exclusively) uses *N*-acylhomoserine lactones (AHL) as signaling molecules. The most common of these AHLs are 3-oxo-hexanonyl homoserine lactone (3-oxo-C6-HSL) and 3-oxo-octanonyl homoserine lactone (3-oxo-C8-HSL) [16].

Natural compounds are attractive candidates for use as antimicrobial agents and virulence inhibitors against pectobacteria [17]. In this context, SA has already been shown to be an excellent antimicrobial candidate, with a wide range of antibacterial and antifungal capabilities [9,18,19]. A treatment that interferes with the growth or virulence of *P. brasiliense* (Pb1692) could be useful for controlling the rot caused by this pathogen during the storage of potato and other tuberous crops.

Here, SA was used as a positive example for the activity of an anti-virulence agent against Pb1692, as we attempted to further increase its antimicrobial potential by chemically modifying the compound. To that end, the potential activity of DSA was predicted by applying computational-docking tools to the QS synthase of *P. brasilense* ExpI, followed by the synthesis of a double-SA composite—bis(2-carboxyphenyl) succinate (DSA)—composed of two salicylic acids conjugated by a succinyl linker. DSA has previously been reported to exhibit anti-inflammatory activity [20] and to control the proliferation of colorectal cancer [21]. Here, DSA was produced and tested for the first time as an antimicrobial agent, revealing antimicrobial activity along with anti-virulence properties.

## 2. Materials and Methods

### 2.1. Chemicals, Bacteria and Growth Conditions

The purities of the commercial bis(2-carboxyphenyl) succinate (Santa Cruz, CA, USA) and salicylic acid (Sigma Aldrich, Rehovot, Israel) used in this work were 98% and 99%, respectively. The stock solutions of both phenolic compounds were prepared in 1% DMSO. Therefore, in all experiments, we included a control treatment with the same volume of 1% DMSO. The strain of the pectobacterium of interest used in this study was *P. brasiliense* Pb1692. *Chromobacterim violaceum* CV026 was cultivated at 28 °C; whereas *E. coli* strains were cultivated at 37  °C. All strains were grown in lysogeny broth (LB) medium (Difco Laboratories, USA) under continuous shaking (150 rpm) in a TU-400 incubator shaker.

### 2.2. Synthesis

Bis(2-carboxyphenyl) succinate [1 eq. of succinyl chloride (109 mL, 1 mmol)] was added drop-wise to a two-neck, 25-mL round-bottom flask that contained 2 eq. salicylic acid (276 mg, 2 mmol). The reaction mixture was stirred at 40 °C overnight. The solvent evaporated and the DSA was purified on a silica gel column using DCM: MeOH, 98:2. The yield was 76%. MS ES + 359. ^1^H NMR (400 MHz, CDCl_3_): δ 7.9 (m, 2H), 7.6 (m, 2H), 7.4 (m, 2H), 7.2 (m, 2H), 2.95 (s, 2H). ^13^C NMR (CDCl_3_, 400 MHz): 170.4, 165.4, 149.9, 133.7, 131.3, 126.1, 123.8, 123.6, 28.8.

### 2.3. Homology Modeling and Docking

A homology model of ExpI (UniProtKB identifier P33882) was generated using the AlfaFold Protein Structure Database. Prior to docking, the model was processed with the Protein Preparation tool in Maestro (Schrödinger, New York, NY, USA), to assign correct protonation states for all residues at physiological pH. SiteMap was used to locate the binding site with 1.026 sitescore and the grid box was centered on the binding site. The structures of the ligands were minimized with the OPLS3e force field and processed using the Ligprep procedure, to assign correct protonation states at physiological pH. The docking of three ligands was modelled: S-adenosyl-L-methionine (SAM), salicylic acid (SA) and bis(2-carboxyphenyl) succinate (DSA). Glide SP (standard precisions) was used for all docking calculations. The binding affinity of each complex was evaluated using IFD scores.

### 2.4. MIC Assay

MIC assays were performed according to Clinical and Laboratory Standards Institute (CLSI) guidelines for determining the antimicrobial activity of the selected phenolic compounds: bis(2-carboxyphenyl) succinate and salicylic acid [22]. Briefly, bacterial cultures of all strains were grown overnight in LB and normalized to 1 × 10^6^ colony-forming units (CFU) mL^−1^ with fresh liquid LB. Twenty μL were then used to inoculate 180 mL of LB containing 2-fold serial dilutions of each of the tested compounds. Cultures of Pb1692 was grown overnight in LB. Cultures were diluted to a final inoculum of 10^6^ CFU/mL in 96-well microtiter plates, compounds were added with appropriate concentrations and MIC was determined after 18 h of incubation at 28 °C. All of the QS assays were performed at concentrations lower than the MIC values (i.e., sub-inhibitory concentrations), to ensure that there was no inhibition.

### 2.5. Biofilm-Inhibition Assay

This assay was performed using the microtiter-dish assay with crystal violet (CV) for biofilm staining, as described by O’Toolee [23]. It was quantified by measuring absorbance at 550 nm in a microplate reader and represented as the absorbance of CV dye bound to biofilm cells. The means of eight replicates were calculated after subtraction of the blank measurement.

### 2.6. Qualitative Assays for the Detection of AHL Molecules

Strain CV026 is a mini-Tn5 mutant of *C. violaceum* in which production of the violet pigment violacein is induced in the presence of AHL compounds with N-acyl C4 to C8 side chains [24]. This assay was performed to assess the effects of bis(2-carboxyphenyl) succinate and salicylic acid on the production of AHLs in Pb1692. The *Pectobacterium* strain was grown in LB overnight as described above. Cultures were then centrifuged (7000× *g*, 5 min at 28 °C) and bacterial pellets were re-suspended in fresh. The reporter strain CV026 was grown in fresh LB medium supplemented with kanamycin (10 μg/mL). Then, bis(2-carboxyphenyl) succinate or salicylic acid was applied to the center of a paper disc on LB plates. The Pc1692 and CV026 were on the plate in concentric circles, with the reporter strain spread a few millimeters away from the tested bacteria. The plates were incubated overnight at 28 °C and the intensity of violet color exhibited by the reporter strain was assessed.

### 2.7. Quantitative Assay for AHL Molecules Using CV026

Quantitative evaluation of QS inhibition of bis(2-carboxyphenyl) succinate and salicylic acid compounds was carried out based on their abilities to inhibit the production of the purple violacein pigment by the CV026 strain, as measured using previously reported methods [25,26]. Pb1692 was cultured in LB for 48 h at 28 °C with or without the addition of sub-MIC concentrations of bis(2-carboxyphenyl) succinate or salicylic acid (0.11 mM and 1 mM, respectively). The cultures were then centrifuged (5223× *g*, 10 min, 4 °C), and the supernatant was sterile-filtered. Five hundred µL of supernatant was mixed with 500 µL of sterile LB inoculated with 5 × 10^6^ CV026 (with kanamycin) and incubated for 24 h at 28°C. One mL of an overnight culture of *C. violaceum* was centrifuged (13,793× *g*, 10 min) to precipitate the insoluble violacein. The culture supernatant was discarded and the pellet was evenly re-suspended in 1 mL of dimethyl sulfoxide (DMSO). The solution was centrifuged (13,793× *g*, 10 min) to remove the cells and the violacein was quantified at a wavelength of 585 nm using a UV-Vis spectrophotometer [SPARK (TEKAN) machine]. The relative level of violacein inhibition was calculated using following the formula: [25]:$$\mathrm{Relative}\;\mathrm{level}\;\mathrm{of}\;\mathrm{violacein}\;\mathrm{inhibition}=\frac{\mathrm{OD}\;\mathrm{of}\;\mathrm{control}\;\mathrm{at}\;585\;\mathrm{nm}-\;\mathrm{OD}\;\mathrm{of}\;\mathrm{test}\;\mathrm{sample}\;\mathrm{at}\;585\;\mathrm{nm}}{\;\mathrm{OD}\;\mathrm{of}\;\mathrm{control}\;\mathrm{at}\;585\mathrm{nm}}\times 100$$

### 2.8. Bioluminescence-Based, Quantitative Assay of AHL Molecules

*E. coli* pSB401 is a bioluminescence-based QS biosensor, which was generated against the background of *E. coli* strain JM109. This strain carries the plasmid pSB401, which possesses the luxRI’:luxCDABE (I’ means luxI-mutated) bioluminescent reporter gene fusion [27]. This system can detect AHLs with acyl chains ranging from six to eight carbons in length (C6 to C8 AHLs) [28]. This strain was used to quantitatively assess the secretion of AHL molecules by Pb1692 in the presence of bis(2-carboxyphenyl) succinate or salicylic acid.

Overnight-grown bacterial pellets were suspended in fresh LB supplemented (or not) with non-inhibitory concentrations of bis(2-carboxyphenyl) succinate or salicylic acid and incubated for 8 h. Control and treated suspensions of Pb1692 were then centrifuged and 10 μL of supernatant was mixed with 190 μL of 5  ×  10^6^ CFU/mL *E. coli* pSB401 in fresh LB medium in 96-well microliter plates. The supernatant used (10 μL) contained bis(2-carboxyphenyl) succinate and salicylic acid and was diluted 20-fold in fresh LB, to a final volume of 200 μL. Two hundred μL of the reporter strain were used as a blank for the experiment and 1% DMSO [carrier of bis(2-carboxyphenyl) succinate and salicylic acid] was used as a control. The plates were incubated at 37 °C for 17 h.

Bioluminescence and optical density were automatically and simultaneously measured every 30 min at 250 ms and 600 nm, respectively, using a Spark^®^ multimode microplate reader (Tecan Trading AG, Switzerland). Bioluminescence was calculated as relative light units (RLU) per unit of optical density at 600 nm, which accounted for the influence of the different treatments on total bioluminescence.

### 2.9. Activities of Hydrolytic Enzymes

The activities of the pectate lyase (Pel), polygalacturonase (Peh) and proteolytic enzymes (Prt) of the *Pectobacteria* strain were tested following overnight exposure of bacteria grown in LB medium at 28 °C to nonlethal concentrations (50% inhibition) of DSA and salicylic acid. Semi-quantitative assays for Peh, Pel and Prt activity were conducted using a plate assay, as described by Chatterjee [29]. The plates were prepared as described and then poked to form 4-mm holes, which were filled with the supernatants from overnight-grown cultures. The plates were then incubated at 28 °C for 24 h. The activities of the enzymes were expressed in terms of the size of the observed haloes. Two independent experiments were carried out, each with four replicates of each concentration of each compound.

### 2.10. Virulence Assays

Virulence was evaluated by assessing symptom severity in two plants—*Zantedeschia aethiopica* (colombe de la paix) and *Solanum tuberosum*, cv. Lady Rosetta—as previously described [30]. Fully expanded, young calla lily leaves and potato tubers were externally disinfected by soaking in 0.7% sodium hypochlorite for 20 min and then washed twice with sterilized distilled water. Whole disinfected potato tubers were inoculated. Discs (20 mm in diameter) were excised from disinfected calla lily leaves and transferred to Petri dishes containing Murashige and Skoog (MS) medium. Bacterial strains were grown overnight in LB liquid medium at 28 °C with continuous shaking and diluted to 1 × 10^7^ CFU mL^−1^ (OD_600_ ¼ 0.1) in sterile DDW containing non-lethal concentrations of the compounds. The bacterial suspensions were shaken at 150 rpm in a TU-400 incubator shaker for 2 h at 28°C before inoculation.

Leaf discs and potato tubers were pierced at the center with a sterile tip and inoculated with 10 mL of bacterial suspension. The inoculated plant material was incubated at 28 °C. For the potato tubers, disease severity was expressed as the percentage of rotten tissue, which was determined at 48 h after inoculation by weighing the whole tuber and then subtracting the corresponding tuber weight after the decayed portion was scraped away. For the calla lily leaves, disease severity was expressed as the percentage of decayed tissue relative to the total area of the disc, at 24 h after inoculation. In both assays, four replicates were performed per strain and compound concentration.

### 2.11. Effects of Bis(2-carboxyphenyl) on AHL Synthesis by E. coli DH5α QS-Negative Strain Complemented with Expi from Pb1692

The QS-negative *E. coli* strain DH5α was used to explore a hypothetical direct interaction of bis(2-carboxyphenyl) succinate with ExpI. To that end, the plasmid pGEM-expI was expressed in DH5α under the control of the T7 promoter [2]. A standard disc-diffusion assay was used to detect the production of AHL using the procedure described above [31]. The DH5α (pGEM) and DH5α/expI + strains were grown in LB overnight, as described above, with an appropriate antibiotic (ampicillin, 100 ug/mL). Cultures were then centrifuged (7000× *g*, 5 min, at 28 °C) and bacterial pellets were re-suspended in fresh LB. The reporter strain CV026 was grown in fresh LB medium supplemented with kanamycin (10 μg/mL).

Then, bis(2-carboxyphenyl) succinate or salicylic acid was applied to the center of a paper disc on an LB plate, with the DH5α (pGEM) or DH5α/ expI+ and CV026 present on the plate in concentric circles, with the reporter strain spread a few millimeters away from the tested bacteria. The plates were incubated overnight at 28 °C and the intensity of violet color exhibited by the reporter strain was assessed.

## 3. Results

A novel method for the synthesis of DSA, a compound that has inconsistent erratic commercial availability, was developed as part of our effort to produce environmentally friendly plant-protection agents. We used a synthesis approach that decreases the damage to the environment by using succinyl chloride to dissolve salicylic acid (Figure 1), thereby avoiding the need for other solvents and reducing environmental damage [32].

### 3.1. Determination of Minimum Inhibitory Concentrations (MICs)

MICs were determined by the broth-dilution method, as recommended by the Clinical & Laboratory Standards Institute, with minor modifications. The MIC was determined as the lowest concentration of the compound at which no measurable growth (i.e., increase in absorbance) occurred for a given strain. MIC values for DSA [bis(2-carboxyphenyl) succinate] are presented in Figure 1a. The MIC value of 1.5 mM DSA was 4-fold lower than the MIC value of 6 mM SA (Figure 1b). Cell counts following 24 h of exposure to the compounds are provided (Appendix A). Further assessments of the effects of the two compounds on growth, motility, biofilm formation, accumulation of signal molecules and Pb1692 infection were carried out using nonlethal concentrations that induced no more than 50% growth inhibition relative to the water-treated control (1 mM or less). In previous work involving the antibiotic ciprofloxacin, these concentrations were shown to have no effect on bacterial virulence [33,34].

### 3.2. Molecular Docking of SAM, SA and DSA to ExpI

Molecular docking was used to determine a mode of antimicrobial action for DSA that results in greater antimicrobial activity. Previously, we proposed a role for SA as a potent inhibitor of AHL synthase and a likely mode of action for that activity [2]. *P. brasiliense*’s QS-signaling molecules—3-oxohexanoyl homoserine lactone (3-oxo-C6HSL) or 3-oxooctanoyl homoserine lactone (3-oxo-C8HSL; AHL)—are biosynthesized from sadenosyl methionine (SAM) and acylated carrier protein by the AHL synthase ExpI, a LuxI ortholog [35].

Analysis of Glide SP docking results showed that SAM forms five hydrogen bonds, two with the side chains of Arg101 and Glu170 and three with the backbone of Ala35 and Ile142. In addition, SAM forms a salt bridge with the backbone of Trp34 (Figure 2a). SA forms one hydrogen bond and a salt bridge with the side chain of Arg101 and has one π–π interaction with the side chain of Phe82 (Figure 2b). DSA forms a salt bridge with the side chain of Arg101, two π–π interactions with Phe82 and Trp34, and two hydrogen bonds with the side chains of Tyr85 and Ser16. The superimposition of SAM, SA and DSA at the binding site of ExpI (presented in Figure 2d) suggests that DSA can compete with the natural effector for binding-site interactions, thereby interfering with the enzyme’s production of AHL.

The substance that best binds to the merged site in ExpI is predicted to be the natural effector SAM (−6.8 kcal mol^−1^), followed by DSA (−5.6 kcal mol^−1^) and SA (−5.0 kcal mol^−1^; Table 1). Using Glide Emodel, we confirmed the lower energy gain associated with the binding of the smaller molecule, SA. Meanwhile, SAM and DSA had ranking values that were close to one another, supporting the predicted higher binding affinity of DSA to ExpI, as compared to SA. These rankings are expected to result in DSA’s impairment of pectobacterial virulence.

### 3.3. Effects of DSA and SA on Motility

Motility is an important virulence trait of many plant-pathogenic bacteria, including *P. brasiliense* [14,36,37]. Here, we compared the effects of DSA and SA on the motility of Pb1692. The highest Pb1692 motility rate was observed for the DDW control treatment. Decreased swimming motility was observed for all nonlethal concentrations of DSA, in comparison to SA (Figure 3). Additionally, 0.6 mM of SA did not have any marked influence on motility; whereas SA concentrations of 1 mM and above were associated with slight, nonsignificant increases in motility relative to the control. Overall, DSA had a stronger inhibitory effect on Pb1692 motility.

### 3.4. Biofilm Formation

Biofilm formation is yet another measure of bacterial virulence. Here, the effect of DSA on the biofilm-formation ability of Pb1692 was assayed using a microtiter-dish assay with non-growth-inhibiting concentrations of the compounds. A significant (*p*  <  0.005) reduction in biofilm formation was observed following exposure to 0.3 or 0.4 mM DSA (Figure 4). SA significantly affected biofilm formation at a concentration of 0.6 and, to a greater extent, at concentrations of 1.0 mM and 1.5 mM. The 1.5 mM SA treatment inhibited growth by about 25%. As both compounds inhibited the growth of *Pectobacterium* in a dose-dependent manner, the only concentrations that could be considered as inhibiting biofilm formation are those that had no effect on bacterial growth.

### 3.5. Effects of DSA on the Production of QS-Signaling Molecules

AHLs are the most common signaling molecules of the QS system of pectobacteria and the main signals that regulate the synthesis of PCWDEs [38]. To gain further insight into the effect of DSA on the QS system, we used two reporter strains to further assess its effect on the production of AHL. *Chromobacterium violaceum* CV026 was used for qualitative and a quantitative assays, based on the production of the purple pigment violacein in the presence of AHL molecules [25]. The exposure of Pb1692 cells to 1 mM DSA inhibited the production of AHL, as observed in terms of a lower level of purple pigment production in the reporter CV026, as compared to the DDW-treated control (Figure 5a). Violacein produced by the reporter was observed in the outermost circle, while the inner circle contained Pb1692, which was exposed to the compounds (DSA or SA) that diffused from the paper disc at the center of the plate.

The results were consistent with the quantitative assay of violacein production that measured the inhibition of violacein production following the treatment of Pb1692 suspensions with DSA or SA. At 0.4 mM, a concentration that did not inhibit growth (by either substance), the inhibitory effect of DSA was 8-fold greater than that of SA (Figure 5b). At 1 mM DSA, Pigment production was inhibited by almost 100%, while growth was inhibited by less than 50%.

The second assay utilized the *E. coli* biosensor strain pSB401, which was generated against the background of *E. coli* strain JM109. This strain carries the plasmid pSB401, which possesses the luxRI′:luxCDABE bioluminescent reporter gene fusion, to quantitatively measure bioluminescence in the presence of AHL-signaling molecules. Here, the reporter strain was used to sense AHL molecules following the exposure of Pb1692 to the compounds [28]. Supernatants of bacterial cultures were pre-exposed to either DSA or SA and bioluminescence levels were correlated with the amount of AHL in the culture for 8 h.

DSA strongly inhibited bioluminescence, as compared to SA or the DDW control (Fifure 6a). The results were completely consistent with those of the first assay, supporting the significant inhibition of AHL production in response to DSA as opposed to SA. Again, at a concentration of 0.4 mM (neither 0.4 mM DSA nor 0.4 mM SA affected growth), the inhibitory effect of DSA on AHL accumulation by Pb1692 was 3-fold greater than that induced by SA. At 0.6 mM, DSA induced 25% growth inhibition, which was 4-fold greater than the growth inhibition induced by the same amount of SA. The growth of the reporter strain was not affected by any of the above treatments (Figure 6b).

### 3.6. Effects of DSA on Exoenzyme Activity

Exoenzyme activity is a virulence determinant that is crucial for necrotrophic pathogens, including pectobacteria. Exoenzymes promote cell-wall degradation and induce soft rot [36,39]. Accordingly, exoenzymes such as pectate lyases (Pel), polygalacturonases (Peh) and proteases (Prt) are expected to play a major role in the inhibition of the virulence of Pb1692. Indeed, Pb1692 exposed to nonlethal concentrations of SA or DSA displayed a greater than 50% reduction in Pel activity, 50% less Peh activity and 40% less Prt activity, according to a semi-quantitative assay. The experiments show that all of the examined concentrations of DSA inhibit PCWDE activity significantly better than all of the examined concentrations of SA. DSA treatment decreased Prt and Peh activity by 45–50%, compared to 20–30% for SA (Figure 7).

### 3.7. Infection following Applications of SA or DSA

We examined the abilities of SA and DSA to control Pb1692 infection in two unrelated hosts and plant tissues (i.e., potato tubers and calla lily leaf discs). Pb1692 was first exposed to nonlethal concentrations of either SA or DSA for 2 h and the cultures were then used to inoculate calla lily leaf discs and potato tubers. In most treatments, the tested concentrations of DSA significantly reduced the ability of the bacterium to induce disease symptoms (Figure 7). This effect could be seen even at the very low concentrations, which had no effect on bacterial growth, and was more pronounced at concentrations that did inhibit bacterial growth. Significant reductions in potato tuber decay were observed following pretreatment of the bacteria with 0.1 or 0.2 mM DSA; whereas SA inhibited infection only at 0.4 mM (Figure 8a), once again demonstrating reduction of disease at a treatment concentration that was insufficient to limit bacterial growth. A 4-fold reduction in the decay of calla leaves was also observed when bacteria were treated with 0.2 mM DSA; whereas a concentration of 0.8 mM SA was required to induce a similar effect (Figure 8b).

### 3.8. Effect of DSA on AHL Biosynthesis in DH5α

The QS-negative *E. coli* strain DH5α was used to explore a hypothetical direct interaction of DSA with ExpI. To that end, the plasmid pGEM-expI was expressed in DH5α under the control of the T7 promoter [2]. Violacein, a purple pigment produced by the reporter strain CV026 in response to AHL, was observed in the outermost circle, while the inner circle contained DH5α/ expI+, which was exposed to DDW (control), SA or DSA that diffused from a paper disc in the center of the dish. Under the control conditions, DH5α/expI+ was able to produce AHL, as shown by the strong synthesis of violacein by the reporter CV026 (Figure 9, lower panel, left). The production of the pigment suggests that the signaling molecule was efficiently produced by the introduction of pGEM-expI to DH5α. Upon treatment with DSA (0.6 mM, 30 µL), the synthesis of AHL by ExpI was almost completely blocked, as indicated by the absence of violacein pigment (Figure 9, top panel, right). Treatment with 1 mM SA had a weaker effect on the inhibition of pigment production. These results are consistent with those of the quantitative assay, demonstrating the direct and strong inhibition of ExpI by DSA.

## 4. Discussion

Efforts to secure adequate food supplies have led to the use of extremely large quantities of pesticides over the past century. The frequent and copious application of these agrochemicals has had numerous deleterious effects on the environment and presented risks to human health. Antimicrobial compounds, whether bactericidal or growth-inhibiting, place strong selective pressures on bacteria to develop resistance. The widespread use of antimicrobial agents has accelerated the emergence of resistant pathogens [40]. In recent years, there has been an ongoing search for new compounds that can be synthesized in quantities sufficient for commercial use. One approach to this has been the re-purposing of compounds whose safety and other properties have already been examined, to effectively control bacterial pathogens [41]. Compounds that have multifunctional control mechanisms (e.g., those that employ a multi-hurdle approach) may be able to overcome the development of resistance traits. The above considerations point to QS as an attractive target for the development of such new pesticides and several anti-QS compounds have been the subject of basic and applied research [3]. Salicylic acid is an excellent lead compound, with well-known antibacterial effects against many Gram-positive and Gram-negative bacteria including pectobacteria [42].

Here, a congener of SA, which has been hypothesized to act as both an inducer of the plants’ own immune system [43] and as an antimicrobial agent, like SA, was examined as a possibly more effective alternative to SA. In line with a previous study, we focused on a congener whose dimensions are relatively similar to those of the natural effector SAM, which we expected would facilitate its binding to ExpI, to impair bacterial virulence rather than killing the bacteria and posing a selective pressure [2].

DSA, a double-SA congener, was selected and we examined its antibacterial capacity by conducting a physiological assessment in a plant-infection model, as well as molecular-docking calculations. DSA was studied in comparison to SA and the activities of the two molecules were evaluated in terms of their abilities to inhibit bacterial growth and to impair virulence.

Our results suggest that DSA is more effective than SA for suppressing pectobacterial infection. As far as we know, neither the antimicrobial activity of DSA nor it ability to impair the virulence of a plant-pathogenic bacterium has been reported previously, as those properties were not the focus of the molecule’s developers. The antimicrobial activity of DSA as a salicylate-based poly(anhydride esters) has been previously reported in the context of *Salmonella enterica*, making its re-purposing promising [44].

The MIC of DSA required to inhibit the growth of Pb1692 was 4 times lower than the MIC of SA. This difference could not be explained by the simple stoichiometry of the two compounds, resulting from the degradation of DSA into its constituents upon treatment. Had this been the case, the MIC value of DSA would have been half that of SA. Nonetheless, based on our MIC determination, the next experiments were calibrated and conducted with sub-MIC–full growth concentrations.

Prior to the performance of more in vitro and in vivo experiments, computational docking was used to select mechanisms of action that may explain the unexpected, lower MIC values. The GlideScore SP values obtained in this work are in agreement with a previously reported study of the docking of C6HSL, SAM and SA to ExpI, which reported GlideScore XP values of −6.4, −6.2 and −5.3 kcal/mol, which translate into relative energies of 0.0, 0.2, and 1.1 kcal/mol for the three ligands, respectively [2]. The results obtained here suggest that DSA may better inhibit QS and hence bacterial virulence, as compared to SA, due to its greater affinity and higher binding rate to the active site of ExpI.

Biofilm formation and motility are known to be tightly controlled by QS in several pathogens, including *P. brasiliense* [14,45,46]. Here, we show that DSA significantly impaired these activities in Pb1692, with DSA exhibiting greater activity than SA at a rate that exceeded the actual concentration of 2-hydroxybenzoic acid (SA) in the suspension (i.e., if DSA had been degraded). For instance, inhibition of biofilm formation was 7-fold greater upon application of 0.4 mM of DSA relative to SA (concentrations that had no effect on growth). Swimming motility was reduced more than 3-fold relative to SA upon application of 0.6 mM DSA and 5-fold upon application of 1.5 mM DSA. These observations were consistent with a clear and significant reduction in the accumulation of AHL in Pb1692 suspensions following exposure to the compounds, as detected by two biosensor strains. The production of violacein by CV026 was reduced in a concentration-dependent manner; whereas SA inhibited pigment production to a lesser extent at the tested concentrations, suggesting that its inhibitory effect on AHL synthesis by ExpI was less effective. The sub-MIC concentrations of DSA repressed AHL production by Pb1692 without restricting bacterial growth, thereby affecting the accumulation and concentration of AHL in the suspension. The same trend was observed for the reporter strain *E. coli* pSB401, which expresses QS response-regulator protein luxR and a bioluminescence cassette [47]. The reporter strain was exposed to AHL produced by Pb1692 following exposure to DSA or SA. DSA strongly reduced the production of AHL and, accordingly, the levels of QS-mediated luminescence in the biosensor strain *E. coli* pSB401.

The stronger effect of DSA (as compared to SA) on QS inhibition had been predicted by computational docking performed at the beginning of this project. However, the results of docking simulations are not always supported by experimental data. Here, our docking simulations have proven to be an efficient tool for predicting the potential of a molecule to inhibit virulence. However, these encouraging results do not rule out the possibility that other mechanisms may be involved in DSA’s antimicrobial and anti-virulence activity against Pb1692.

## 5. Conclusions

Our findings show that DSA inhibits bacterial growth and virulence. These effects of DSA are stronger than those of SA, which has been previously shown to inhibit the virulence of several bacterial species [48,49]. Moreover, DSA had a significant effect on several virulence determinants at concentrations that did not impair bacterial growth. A comparison of that anti-virulence activity with the anti-virulence activity of SA revealed that this effect could not be attributed to the simple stoichiometry of the two compounds. Our findings suggest that DSA’s activity may be due to its greater affinity for the active site of ExpI, as suggested by the docking results. Alternatively, it may be due to some other mechanism that has yet to be discovered. In any of these scenarios, DSA holds great potential as an anti-virulence agent with broader applications.

## Data Availability

The data sets of this study are available on reasonable request from the corresponding authors.

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
