# Peer review of "Disalicylic Acid Provides Effective Control of Pectobacterium brasiliense"

_microorganisms, 2022, doi:10.3390/microorganisms10122516_

Round 1

Reviewer 1 Report

This manuscript presented the inhibitory effect of disalicylic acid on Pectobacterium brasiliense. MIC, biofilm formation, secretion of cell-wall degrading enzymes, and motility and production of AHL were analyzed. The research is meaningful and the methods are correct. I think it could be accepted by this journal after major revision.

Major concerns:

1. Scheme 1: It’s suggested to present the synthesis reaction equation. That’ll be more helpful for readers.

2. The spacing of the x-coordinate values is uneven. It is recommended to display the data in the form of a table.

3. Figure 4: Which column represents DSA, and which is SA? The authors should label clearly.

Author Response

Q: Scheme 1: It’s suggested to present the synthesis reaction equation. That’ll be more helpful for readers.

A: Thank you for the comment, we have modified Scheme 1 accordingly to describe a one-step synthesis.

Q: The spacing of the x-coordinate values is uneven. It is recommended to display the data in the form of a table.

A: Thank you for the comment. The x-coordiante has been corrected to present the true values.

Q: Figure 4: Which column represents DSA, and which is SA? The authors should label clearly.

A: Following the reviewer’s comment, we have added the missing label to Figure 4.

Reviewer 2 Report

Dear colleagues,

The results obtained in your study are extensive and demonstrate the effectiveness of Bis(2-carboxyphenyl succinate) in combating P. brasiliense. How will it be applied in culture and what are the biodegradation products? Remaining in soil and plants.

I did not understand if in the experiments you used commercial bis(2-carboxyphenyl) succinate or the synthesized one. What solvent was removed after synthesis, if you added succinyl chloride over salicylic acid and refluxed at 40 or 50 degrees and you stated that the use of solvent was avoided. Then what refluxed at that temperature? Characteristics of the synthesized compound

Author Response

Q: How will it be applied in culture and what are the biodegradation products? Remaining in soil and plants.

A: Thank you for the comment.  We have removed the word biodegradable from the abstract, as we do not know the full spectrum of biodegradation products.

Q: I did not understand if in the experiments you used commercial bis(2-carboxyphenyl) succinate or the synthesized one.

A: Since the material is unstable, we chose to use the ‘in house’ synthesized compound.

Q: What solvent was removed after synthesis, if you added succinyl chloride over salicylic acid and refluxed at 40 or 50 degrees and you stated that the use of solvent was avoided. Then what refluxed at that temperature?

A: Using succinyl chloride, a liquid at the temperature range used here, as both a reagent and a solvent would reduce environmental pollution by reducing the need for additional solvents.

To clarify we have added the annotation “succinyl chloride (l)” above the reaction arrow in scheme 1. We have also changed the use of “mg” in the text and we now use “ml” to detail the amount of succinyl chloride used.

Q: Characteristics of the synthesized compound.

A: We have added the analytical characteristics of the synthesized compound to the text: Mass spectrometry data, proton NMR and carbon NMR of the synthesized DSA (synthesis in experimental section):

“MS ES+ 359. 1H NMR (400 MHz, CDCl3): δ 7.9 (m, 2H), 7.6 (m, 2H), 7.4 (m, 2H), 7.2 (m, 2H), 2.95 (s, 2H). 13C NMR (CDCl3, 400MHz): 170.4, 165.4, 149.9, 133.7, 131.3, 126.1, 123.8, 123.6, 28.8.”

Round 2

Reviewer 1 Report

The authors have revised careflly according to the reviewer's comments and suggestions. I think it could be accepted by this journal.